# UK cancer vaccine advance – Recognising and realising opportunities

Charles Craddock[1], Philip Earwaker[2], Matthew Fittall[3], Elisa Fontana[4], Divya Ganesh[5], Marco Gerlinger[6], Qamar Ghafoor[7], Robert P Jones[8], Victoria Kunene[7], Lennard Lee[5] , Rebecca Lee[9], Siow-Ming Lee[10], Mark Linch[10], Martin Little[11], Justin Liu[12], Hayley McKenzie[13], Russell Petty[14], David J Pinato[15,16,17], Thomas Powles[6], Andrew Protheroe[5], Tim Robinson[18], Paul J Ross[19], Kai Keen Shiu[10], James Spicer[20], Stefan Symeonides[21], Michael Tilby[22], Dale Vimalachandran[23], Jenny Y Wang[5], Andrew Wardley[24] and Helen Winter[25]

[1]University of Warwick Warwick Clinical Trials Unit, Coventry, United Kingdom of Great Britain and Northern Ireland; [2]Cambridge University Hospitals NHS Foundation Trust, Cambridge, United Kingdom of Great Britain and Northern Ireland; [3]University College London, London, United Kingdom of Great Britain and Northern Ireland; [4]Sarah Cannon Research UK, London, United Kingdom of Great Britain and Northern Ireland; [5]University of Oxford, Oxford, United Kingdom of Great Britain and Northern Ireland; [6]Barts Health NHS Trust, London, United Kingdom of Great Britain and Northern Ireland; [7]University Hospitals Birmingham NHS Foundation Trust, Birmingham, United Kingdom of Great Britain and Northern Ireland; [8]University of Liverpool, Liverpool, United Kingdom of Great Britain and Northern Ireland; [9]The Christie NHS Foundation Trust, Manchester, United Kingdom of Great Britain and Northern Ireland; [10]University College London Hospitals NHS Foundation Trust, London, United Kingdom of Great Britain and Northern Ireland; [11]Oxford University Hospitals NHS Foundation Trust, Oxford, United Kingdom of Great Britain and Northern Ireland; [12]Leeds Teaching Hospitals NHS Trust, Leeds, United Kingdom of Great Britain and Northern Ireland; [13]University Hospital Southampton NHS Foundation Trust, Southampton, United Kingdom of Great Britain and Northern Ireland; [14]University of Dundee, Dundee, United Kingdom of Great Britain and Northern Ireland; [15]Imperial College London, London, United Kingdom of Great Britain and Northern Ireland; [16]Hammersmith Hospitals NHS Trust, London, United Kingdom of Great Britain and Northern Ireland; [17]Universita degli Studi del Piemonte Orientale Amedeo Avogadro, Vercelli, Italy; [18]University of Bristol, Bristol, United Kingdom of Great Britain and Northern Ireland; [19]Guy's and St Thomas' NHS Foundation Trust, London, United Kingdom of Great Britain and Northern Ireland; [20]King's College London, London, United Kingdom of Great Britain and Northern Ireland; [21]Edinburgh Cancer Research Centre, Edinburgh, United Kingdom of Great Britain and Northern Ireland; [22]University Hospitals Coventry and Warwickshire NHS Trust, Coventry, United Kingdom of Great Britain and Northern Ireland; [23]Countess of Chester Hospital NHS Foundation Trust, Chester, United Kingdom of Great Britain and Northern Ireland; [24]Outreach Research & Innovation Group Limited, United Kingdom of Great Britain and Northern Ireland and [25]University Hospitals Bristol NHS Foundation Trust, Bristol, United Kingdom of Great Britain and Northern Ireland

## Comment

**Keywords:**
Cancer; Clinical trials; Immunotherapeutics; Personalized therapies; Precision oncology

**Corresponding author:**
Lennard Lee;
Email: lennard.lee@nhs.net



## Abstract

Vaccines have revolutionised the field of medicine, eradicating and controlling many diseases. Recent pandemic vaccine successes have highlighted the accelerated pace of vaccine development and deployment. Leveraging this momentum, attention has shifted to cancer vaccines and personalised cancer vaccines, aimed at targeting individual tumour-specific abnormalities. The UK, now regarded for its vaccine capabilities, is an ideal nation for pioneering cancer vaccine trials. This article convened experts to share insights and approaches to navigate the challenges of cancer vaccine development with personalised or precision cancer vaccines, as well as fixed vaccines. Emphasising partnership and proactive strategies, this article outlines the ambition to harness national and local system capabilities in the UK; to work in collaboration with potential pharmaceutic partners; and to seize the opportunity to deliver the pace for rapid advances in cancer vaccine technology.

## Impact statement

Cancer vaccines and personalised cancer vaccines represent a potential paradigm shift in oncological care. However, the technology is not without its pitfalls and requires a package of solutions to be able to flourish. This article is the first national leadership strategy report into the cancer vaccine advance and the opportunities for those affected by cancer.

## Introduction

Vaccines have been one of the most important therapeutic advances in medicine, resulting in the eradication of many infectious diseases such as smallpox, and in the control and near-eradication of polio. In recent years, we have again experienced their transformative benefits, with multiple vaccines effective in preventing illness from SARS-CoV-2 (Monin et al. 2021; Falsey et al. 2021). Furthermore, pandemic experience has demonstrated an extremely rapid benchmark for vaccine development. Rather than years or decades, implementation can be within months, given sufficient awareness and prioritisation of pre-clinical studies, financing, manufacturing, distribution, regulatory approvals, system delivery, and product updates (Lurie et al. 2020). Consequently, there is currently strong public, scientific, and clinical support for vaccine studies.

In the pursuit of new applications for vaccine technology, attention has turned towards another potentially transformative frontier: vaccines against cancer. The aim is to harness technologies and processes utilised during the pandemic to develop a vaccine that can direct the body's immune response against cancer cells. During the last decade, oncology has been transformed by immunotherapies entering the mainstream, specifically immune checkpoint inhibitors (Darvin et al. 2018) and cell therapies like CAR-T and effector T-cell therapy (Rallis et al. 2021). These have led to long-term disease control, immunity, and even cures for some patients with advanced cancer, and current research is increasingly focusing on earlier lines of therapy, including adjuvant approvals. However, checkpoint inhibitors are largely unguided broad-brush approaches, reducing immunosuppressive signals rather than focussing immunity against detected cancer-specific abnormalities.

In recent years, the United Kingdom has provided major contributions to progress in effectively developing and deploying vaccines at scale, developing the Oxford-AstraZeneca SARS-CoV-2 vaccine ('Oxford Vaccine Saved Most Lives in Its First Year of Rollout | University of Oxford' 2022) and being the first to administer the Pfizer-BioNTech SARS-CoV-2 vaccine outside a clinical trial (Page 2020). This recent experience and its associated academic, clinical, and regulatory infrastructure should make the UK a very promising location for the pivot to cancer vaccine trials. Indeed, the UK government has signalled intent with a target to enrol 10,000 patients into vaccine trials for cancer across multiple different subtypes and stages by 2030, and to stimulate a global technology advance ('UK–BioNTech Partnership for mRNA Cancer Vaccines–The Lancet Oncology', n.d.).

Vaccines against cancer can be fixed (termed "off the shelf"), where everyone receives the same vaccine, and thus benefit from being mass manufactured. Or they may be individualised to the abnormalities seen in an individual patient's cancer, an approach that may be the next major disruptive therapeutic modality (Lin et al. 2022). Historically, progress in developing cancer vaccines has been limited, and significant clinical impact has not yet been achieved. However, the outlook has changed. Firstly, new technologies like mRNA have achieved widespread acceptability. This means that it is easier than ever to update vaccines. Secondly, our understanding of cancer genomics has been transformed. We now have a better understanding than ever of the antigens that are observed in cancer. Thirdly, vaccines are now being pioneered in a new clinical sphere, in earlier stage cancers to prevent recurrence. Finally, vaccines can be made personalised to the individual changes seen in each cancer, thus heralding the reality of true precision medicine approaches.

Cancer therapies do pose significant additional challenges compared to developing COVID-19 vaccines. Firstly, COVID-19, as a perceived existential threat, became the sole political, economic, academic, and healthcare priority of the time. Cancer research, rightly, must compete for resources amidst other important societal needs. Secondly, cancer is a much more complex and variable disease than COVID-19.

This article brings together leading clinicians and scientists across the UK to explore obstacles, opportunities, and solutions to cancer vaccine development. We shed light on the multifaceted issues encountered along each step of the journey: from trial design and the establishment of a trial delivery system, through to the engagement of hospitals, patients, and researchers. A partnership model is essential, and we therefore discuss the collaborations needed amongst national bodies, local systems, and commercial partners. We hope to initiate the dialogue for a collaborative and proactive strategy to deliver development.

## National systems

The United Kingdom National Health Service still commands widespread confidence and support from the public and its centralised systems may be an asset to large-scale trial delivery ('Trust in Government, UK–Office for National Statistics', n.d.).

Awareness and effective communication are critical. Across the board, awareness and support must be maintained, including within the public, patient groups, charities, primary care and specialist centres. Public confidence can be easily lost, particularly when commercial partners are involved. Previous successes, such as the 100,000 Genomes Project, are examples of patient engagement done generally well and at scale. Awareness could be achieved by appointing an accountable leader who will establish and maintain momentum. National infrastructure may need to be coordinated and restructured to better streamline delivery, for example, regulatory bodies must be given assistance and support to ensure that they can achieve their domestic UK mission "of approving all clinical trials within 60 days" ('What We're Doing to Speed up Clinical Trials in the UK – Department of Health and Social Care Media Centre' 2023). Reducing duplication of documentation and processes may result from the development of a unified review system. This is particularly relevant as cancer vaccine trials, in contrast to COVID-19 mRNA vaccines, are currently considered as advanced therapeutic medicinal products (ATMPs). This imposes extra oversight at a local and national level and could be alleviated by a single national or regional process.

Trials can also be facilitated through systems that achieve universal opportunity to participate, focussing on underrepresented groups, thus maximising enrolment by breaking down geographical, financial, and cultural barriers. This will require community engagement and empowerment policy programmes, and targeted interventions to address the specific needs and concerns of these populations.

Using national data systems, leveraging electronic health records (EHR) to screen eligible participants with virtual/remote consent may avoid the natural biases towards the academic centres. This would be in keeping with aspirations to reduce inequalities, such as England's CORE20PLUS5 initiative ('NHS England » Core20PLUS5 (Adults)–an Approach to Reducing Healthcare Inequalities', n.d.) and Scotland's Equity of Access initiative ('Improving Equity of Access to Cancer Clinical Trials in Scotland', n.d.). In the long-term, using routine national data may enhance outcome monitoring in 'real world settings' at much reduced cost. This could

be a universal offering to patients, with an associated total redesign of the patient research journey (Inan et al. 2020), while also maintaining patients' autonomy to "opt out" from research participation.

The advanced technologies required for vaccine development may be better suited to nationally coordinated infrastructure. Regional genomics laboratories may centralise some of these techniques, but they will need to work together to standardise approaches and maintain access to routine and rapid tumour sequencing, required for tumour-specific target antigen identification. This could include harmonising activities across Scotland, Northern Ireland, and Wales, and potentially introducing key performance indicators. Genomics laboratories need to invest, as liquid biopsy genomics technologies are transforming the diagnostic landscape. Many cancer vaccines are targeted towards specific antigens, which rely on markers detected via immunohistochemistry, RNA or, more commonly, DNA mutations detected via genomic profiling platforms that are progressing from selective NGS panels to broad whole genome approaches. The use of a central or partner-mandated companion diagnostic can introduce cost and delay. Thus, this should be addressed, particularly in the case of personalised vaccines, where profiling is intertwined with the vaccine identity, and in the case of adjuvant trials, where detection of circulating free DNA is increasingly used for patient selection. Potential solutions include empowering local systems to establish reflex testing and a nationwide/regional/local programme of broad profiling that could be done with leading academic institutions, recognising the need for pathology and workforce buy-in and resourcing. This may include tumour and HLA genetics and expanding research infrastructure to accommodate future, more detailed serological immune assays and/or frozen tissue samples. The benefits and synergies for other cancer research and therapy development are clearly apparent.

If achieved, national reform of research systems will have modernised the sector, delivering a tempo of trial achievements similar to during the pandemic response. This in turn would create jobs, upskill the workforce, draw further investment, give new treatment opportunities for patients, and ultimately enable the advance of vaccines for cancer.

### Local systems

The efficient rollout of cancer vaccine research and future incorporation into clinical practice will be underpinned by NHS cancer centres. It is important to ensure that centres are supported and empowered with the autonomy to move agilely.

In the short-term, firstly, local research infrastructure must have sufficient staff, resources, and funding to deliver trials. The Royal College of Radiologists' latest census revealed that the UK had a 15% shortfall of clinical oncologists in 2021, a figure which is projected to escalate to 25% by 2027 ('RCR Clinical Oncology Workforce Census 2022 | The Royal College of Radiologists', n.d.). This issue also extends to medical oncologists, particularly in smaller hospitals and district general hospitals. Last year, 1 in 5 medical and clinical oncology training posts were unfilled ('Specialty Recruitment: Round 1–Acceptance and Fill Rate | Health Education England', n.d.). Workforce upskilling, training and education issues must be addressed to ensure adequate execution of cancer trials. The ongoing upskilling and training and education of staff could be achieved through specialised research training programmes. For clinicians such as surgeons or oncologists, research must be a viable and appealing sub-specialisation career pathway, while involvement in research must also be embedded as core to every clinician's practice. If resourcing is perceived as

an issue, there must be acknowledgement that clinical trials require an associated investment into staff time and often into training and development opportunities. Time spent by staff in activities dedicated to the cancer vaccine advance must be acknowledged and ring-fenced within job plans to ensure that focus is maintained. This can be specifically included in clinical role programmed activities (PA) allocations. Recognition of staff and patients participating in this research should be maximised to boost morale and mitigate against 'research fatigue'.

One method of expanding resourcing to support trial infrastructure is to ensure that gains obtained from cancer vaccine research should be reinvested into teams directly delivering these initiatives, and this would be in keeping with a recent independent report into the "future of commercial clinical trials" by Lord James O'Shaughnessy ('Commercial Clinical Trials in the UK: The Lord O'Shaughnessy Review–Final Report', n.d.). By giving more autonomy to these research teams to manage their resourcing and funding, this would act to empower the local system to seek more opportunity for cancer vaccine trials. It may function as a franchise model with investment ploughed back into research units, who could seek out more long-term partnerships and opportunities. These reforms could lead to streamlined systems capable of opening sites rapidly and delivering to ambitious recruitment targets. This need not be at the expense of nationwide patient access to research; instead successful centres with the resources to train and support other up-and-coming centres would allow the most successful research models to spread across the country.

Trials should be designed in such a way that they are easy to incorporate into existing clinical healthcare pathways. Local healthcare systems can act proactively to ensure that there is sufficient capacity in local diagnostic teams such as radiology, pathology, and genomic labs. Surgical teams can be empowered to ensure that sites approach peri-operative patients early, and that sampling requirements for molecular profiling are met and of sufficient quality: be they fresh or FFPE samples, or ctDNA from blood. In addition, local clinical IT systems should be equipped with fail-safes to support uninterrupted cancer vaccine research delivery, ensuring that new electronic healthcare system upgrades do not halt the momentum for this research field. Finally, local healthcare systems should consider establishing cancer vaccine boards that report to the trust executive, to ensure proactive communication with research leads. These boards could also facilitate engagement with potential principal investigators, drive recruitment, and maximise expertise in the field.

In parallel, an uplift project must be started whereby the footprint of local cancer centres capable of delivering cancer vaccines should be comprehensively expanded. There is no reason that small cancer centres, often located geographically away from academic institutions or major hospitals, should not be permitted to conduct or considered for cancer vaccine research. An uplift programme that enables research-naïve centres to conduct such studies will enable cancer vaccine trials to reach more people and achieve universal research provision and equity of access. Local systems willing to advance vaccines for cancer within their centres should gain acknowledgement of their ability and achievement by being recognised as "cancer vaccine delivery centres" or "cancer vaccine centres of excellence". Demonstration of their ability to process documentation, including contractual agreements, ethics, and review in a timely manner might be a suitable key performance indicator that defines levels of capability. Additional metrics could involve patient recruitment metrics, quality, as well as safety.

If an uplift programme is not possible, there must be clear mapping of research referral patterns. These should be reinforced,

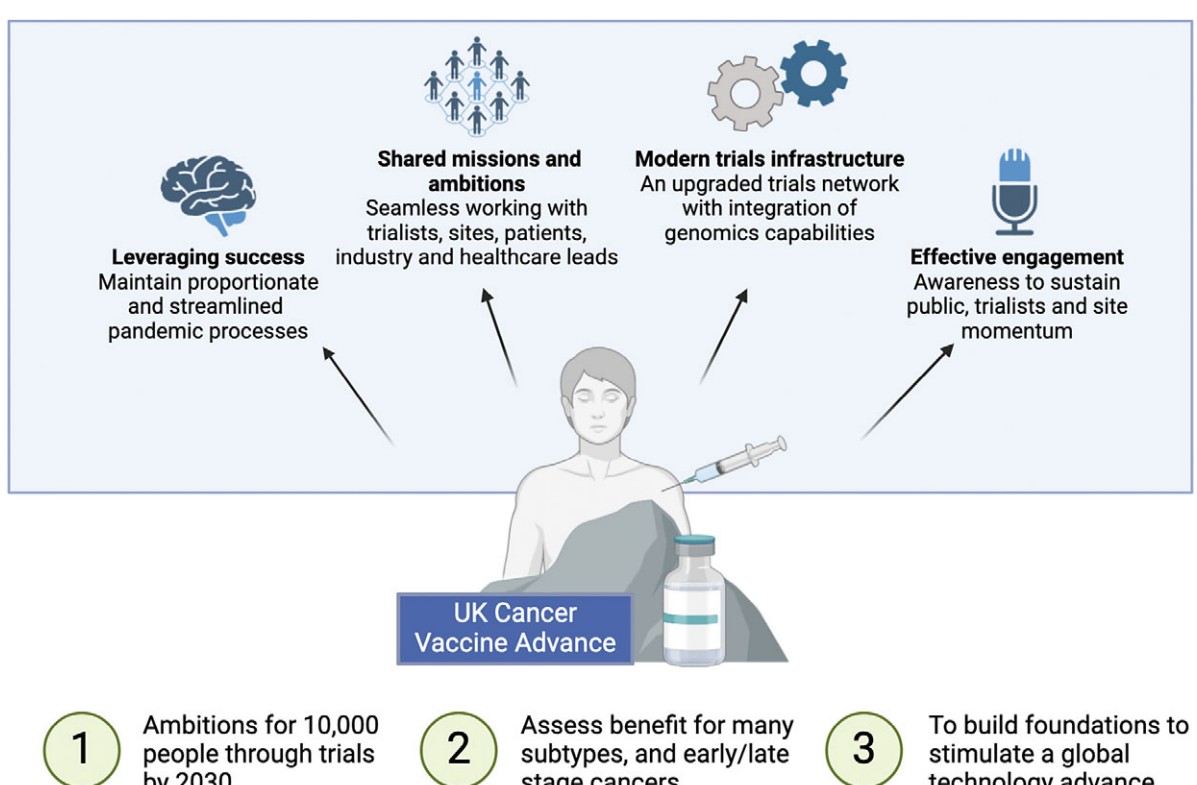

**Figure 1.** Solutions that if achieved by local systems, national systems, and pharmaceutical partners will help facilitate a world-leading UK Cancer Vaccine Advance.

**Figure 2.** Graphical abstract which depicts an overview of the UK Cancer Vaccine Advance.

such that these research networks would function to drive recruitment, minimise patient travel and inconvenience, and yet enable effective regional cooperation and awareness.

## Pharmaceutical partners

Delivery of new therapies ultimately requires commercial collaboration. Pharmaceutical partners must be integral to plans to create an environment of mutual co-operation that will enable cancer vaccines to succeed. The current re-imbursement model for research activity, rather than an up-front investment model, may need to be reviewed, as it leads to a significant lag in capacity building. An upfront investment model with suitable contractual terms could act to reassure both sides, particularly pharmaceutical partners of delivery, whilst simultaneously ensuring that trial processes are streamlined from the start. Upfront resourcing can then be invested to support required background staff, including research nurses and data managers, and better guarantee that a cancer vaccine advance vehicle will hit full speed from the first day of trial opening. Furthermore, sites that recruit the most patients should be further incentivised, by being able to appoint more clinical and backroom staff to build greater capabilities.

Optimising site efficiency and enhancing collaborations can be effective tools from pharmaceutical partners. Monitoring 'conversion' rates for trials could be a shared activity that helps to improve studies. Some studies that recruit poorly may end up being easier to deliver in other countries. Mandated on-site physical health and safety mandates by sponsors or pharmaceutical partners should be carefully considered and updated based on evidence, for example, the observation period of patient's post-injections/infusions. Whilst extensive safety checks and processes may de-risk trials for sponsors or pharmaceutical partners, a pragmatic level of safety checks and process requirements will enable systems to deliver trials more effectively. Structurally, it could be better to place trials with a regional strategy, for example, allowing regions to lead on cancer vaccines for particular tumour indications, as this may mitigate competition between principal investigators and companies, promoting collaboration.

In the longer-term, commercial support for decentralisation of trial activity outside the physical footprint of a hospital is a concept, possible in the United Kingdom, that would build capacity for trial activities. This would reduce the burden on patients to travel long distances and better incorporate trial activity into daily life without mandating multiple trips to cancer centres. This could include community-based blood sampling, mobile apps to collect outcome data, community provision of radiology, or vaccine delivery closer to a patient's home through mobile infrastructure. These may involve new contracts with community, primary care, or independent sector vaccine delivery partners and could achieve improved patient outreach and experience.

When looking to accelerate regulatory approval for cancer vaccine trials, lessons can be drawn from our domestic success during the pandemic, when our regulators, the Medicines Health Regulatory Agency (MHRA), took an innovative and flexible approach, with an expedited review process for COVID-related trials. Working closely with regulatory agencies early on will ensure that studies not only meet regulatory requirements but are done with the strongest level of support by key groups and champions. Two-way conversations allow trials to be further tailored and adapted to meet domestic regulatory standards. This could include utilising the Innovative Licensing and Access Pathway 'Innovation Passport', which allows for MHRA regulatory input into trial design

from the very start. The regulatory pathway of cancer vaccines may even run in parallel to other ATMPs or cancer therapies, given that their safety profile is often higher than conventional cancer therapies.

Deep partnerships with translational research bodies will undoubtedly forge the path for a continual, iterative process of cancer vaccine improvement. Long-term goals must surely involve improving the accuracy of tumour neoantigen identification, incorporating feedback of failed targets from lack of immunological or clinical response, and realising the potential of concepts such as immunobridging. Partnerships with research bodies can allow for the identification and establishment of translational infrastructure for innovative pharmacodynamic, molecular, or genomic surrogate trial endpoints, potentially capturing T-cell phenotype, activation and infiltration. These can include the development of immune assays that may require rapid sample delivery and handling at experienced centres. Deeper investment in translational research will reap dividends in identifying relevant predictive biomarkers for patients' responses to cancer vaccines, which then allow for ever more precision approaches. Investment should also be concentrated towards increasing vaccine manufacturing capabilities.

## Summary of main recommendations

The article is the first national leadership strategy report into the cancer vaccine advance. Significant progress is being achieved through a multi-faceted approach, one that removes barriers to the cancer vaccine advance, and empowers systems. A few key recommendations have been highlighted where the potential gain is most significant (Figures 1 and 2).

1. **Leverage COVID-19 Vaccine Success.** Insights can be drawn from the rapid development and deployment of COVID-19 vaccines. Processes were streamlined, pragmatism prevailed over perfections and groups were able to make timely decisions. Cancer vaccine trials should be delivered in a similar fashion.
2. **Create shared missions.** There is a multitude of pharmaceutical partners with new cancer vaccine technologies against different subtypes of cancers. Everyone should be open to facilitating ambitious long-term partnerships between trialists, sites, industry and healthcare leads to forge the path for continual iterative improvement of cancer vaccine technologies.
3. **Modern trials infrastructure.** Cancer vaccines requires effective trials infrastructure with seamless integration of genomic data across different platforms and institutions. Trial infrastructure and teams should receive investments to upgrade and be made as effective as possible.
4. **Engage public and trialists.** Greater attempts can be made at fostering effective public awareness. This includes understanding the potential opportunities and limitations of cancer vaccine technology. We should aim to garner support and increase the momentum of clinical trials at small as well as large cancer centres.

## Conclusions

The United Kingdom provides a good example of a fertile vaccine research environment that can be supported to deliver global developments in cancer vaccines. The country's rapid and coordinated pandemic response to deliver SARS-CoV-2 vaccine trials has highlighted strengths in academic research, translating genomics from bench-to-bedside, running fast but comprehensive

trials, showcasing vaccine safety and efficacy around the country, and delivering rapid innovation within a single healthcare system. Proactive engagement of trial infrastructure, awareness at a national and local level, and provision of information to the public are key to recruitment. National and local partnerships are critical to ensure sufficient research and clinical capacity to accommodate trials. The silver lining of difficult recent years could be that the environment has been created for harnessing the UK's national and local systems, along with potential pharmaceutic partners, to seize the opportunity and deliver the pace for rapid advance in cancer vaccine technology. This has the potential to transform the lives of the global 50% who will experience cancer during their lifetime, making up hundreds of millions living with malignant disease.

**Open peer review.**  To view the open peer review materials for this article, please visit http://doi.org/10.1017/pcm.2024.5.

**Author contribution.**  All authors contributed significantly to the conception, design, execution, and interpretation of the paper. Each author has reviewed and approved the final version of the manuscript. The contributions of each author comply with the guidelines set forth by the International Committee of Medical Journal Editors (ICJME). All authors have agreed to be accountable for all aspects of the work, ensuring its accuracy and integrity, with the responsibility of the final decision to submit held by corresponding author, Dr. Lennard YW Lee.

**Financial support.**  All expenses associated with the study, including review, and publication, were covered by the authors themselves. This research was conducted without any external financial support or funding.

**Competing interest.**  None of the authors have any conflict of interest relevant to this article.

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
