## [Reviewer Report]

This is a nice review: concise and easy to read with myriad insightful suggestions.

GENERAL COMMENTS

1. The review would benefit from a summary of the specific conclusions and recommendations made through the article, perhaps as a numbered list at the end, pulling together the key areas covered (patient engagement, genomics infrastructure, regulatory, commercial partnerships etc.)

2. It would be helpful if the authors clarified from the start exactly what type of cancer vaccines they are considering. Overall, the article feels like it is focussed on personalised vaccines; indeed, this is where the real interest and challenges are. However, the reference in line 80 to immunohistochemistry is relevant mainly to off-the-shelf vaccines that are more straightforward and perhaps not the focus of this review.

3. Cancer vaccines have been around for a long time without making a significant clinical impact; please could the authors add some thoughts as to why now is the right time to re-explore this approach at scale.

MINOR COMMENTS (for consideration only)

1. Polio has not quite been eradicated yet (but smallpox has). I think perhaps that smallpox is the only infectious disease that has been completely eradicated by vaccination, although this is not to underplay to massive importance of vaccination to the control of infectious diseases.

2. The section on patient engagement is good and a really important area to highlight. Consider noting the care.data debacle as an example of how quickly public confidence can be lost, especially when commercial partners are involved. Also the 100K genome project as an example of patient engagement generally done well and at scale.

3. Decentralisation of clinical trials is another interesting area raised in this review; perhaps a few lines on how this could be achieved, for example using mobile apps to collect outcome data. Also a note on how under-reached groups can be encouraged to participate (another important area rightly raised by the authors).

4. Running personalised vaccine trials at scale will put a strain on the GLHs and will likely require additional resources and new technologies, including analysis of ctDNA as noted by the authors along with whole exome sequencing which is currently not offered. Two more key issues with the current provision of genomics in the UK is the lack of key performance indicators and the lack of harmonisation with (and often funding for) Scotland, N Ireland and Wales.

5. For NHS clinicians, clinical trial activity is currently not generally included in job plans or accounted for in PA allocation. The authors allude to this; a more specific recommendation might be helpful here.

---

## [Reviewer Report]

All comments addressed. I very much enjoyed reading this; it is a valuble contribution to the field.